# Epidural Injection Method for Long-Term Pain Management in Rats with Spinal Stenosis

**DOI:** 10.3390/biomedicines11051390

**Published:** 2023-05-08

**Authors:** Jin Young Hong, Hyunseong Kim, Junseon Lee, Wan-Jin Jeon, Changhwan Yeo, Hyun Kim, Yoon Jae Lee, In-Hyuk Ha

**Affiliations:** Jaseng Spine and Joint Research Institute, Jaseng Medical Foundation, Seoul 135-896, Republic of Korea; vrt23@jaseng.org (J.Y.H.); biology@jaseng.org (H.K.); excikind@jaseng.org (J.L.); cool2305@jaseng.org (W.-J.J.); duelf2@jaseng.org (C.Y.); khyeon94@jaseng.org (H.K.); goodsmile@jaseng.org (Y.J.L.)

**Keywords:** epidural injection, spinal stenosis, low back pain, safety

## Abstract

Epidural injection is one of the most common nonsurgical treatment options for long-term pain relief in lumbar spinal stenosis. Recently, various nerve block injections have been used for pain management. Among them, nerve block through epidural injection is a safe and effective method for the clinical treatment of low back or lower extremity pain. Although the epidural injection method has a long history, the effectiveness of long-term epidural injections in disc diseases has not been scientifically proven. In particular, to verify the safety and efficacy of drugs in preclinical studies, the route and method of drug administration in terms of the clinical application method and duration of use must be established. However, there is no standardized method for long-term epidural injections in a rat model of stenosis to identify the precise efficacy and safety of epidural injections. Therefore, standardizing the epidural injection method is very important for evaluating the efficacy and safety of drugs used for back or lower extremity pain. We describe the first standardized long-term epidural injection method for evaluating the efficacy and safety of drugs according to their route of administration in rats with lumbar spinal stenosis.

## 1. Introduction 

Epidural injection treatment for disc disease is a type of therapy that involves injecting medication into the epidural space of the spinal canal. The goal of this treatment is to reduce pain and inflammation caused by a herniated disc, spinal stenosis, or other conditions that affect the nerves in the spine in order to provide long-term pain relief for patients [1,2]. The procedure involves the injection of substances, such as a combination of a local anesthetic and a corticosteroid, into the epidural space located between the dura mater and the vertebral canal that surrounds the spinal cord [3]. Overall, it is generally considered a safe and effective option for many patients with chronic pain, but some researchers have questioned its long-term effectiveness, and others have raised concerns about its potential side effects, such as infections, bleeding, and nerve damage [4,5]. In addition, scientific evidence regarding epidural injection treatment for disc disease is lacking. In particular, there is a lack of consistency in the methods of long-term epidural administration used in preclinical research, which has led to inconsistent results, making it difficult to compare studies.

In preclinical trials, the method and route of drug administration are critical factors that affect the success of the trial and the validity of the results [6]. The chosen method and route of administration can affect the pharmacokinetics (i.e., how the drug is absorbed, distributed, metabolized, and eliminated from the body) and pharmacodynamics (i.e., the biological effect of the drug) of the tested drug, which in turn can influence its safety and efficacy [7,8]. Several studies have used the rat epidural administration method to investigate various aspects of pain and anesthesia, including the mechanisms of action of different drugs and the effects of different dosages and administration routes [9,10,11]. The rat epidural administration method has been described in several experimental studies.

One such study by Kim et al. described the use of the percutaneous transforaminal epidural injection method in rats using X-rays for the delivery of pharmacological studies and pain management to the spinal epidural space [12]. Although this newly proposed method is minimally invasive to the experimental rat, it rarely causes delivery failure and complications; rats and experimenters have to be exposed to radiation, which cannot be performed without an X-ray generator. Most other methods of epidural catheterization have been performed by inserting a polyethylene tube (PE-10) caudally into the epidural space to position the catheter tip at the drug injection site [11,13,14,15]. However, these methods generally allow a single administration of the drug owing to infection problems caused by exposure to the external end of the polyethylene tube. Previous studies have used implantable mini-osmotic pumps to continuously deliver drugs into the intrathecal space [16]. An osmotic pump (Alzet, Alza, Palo Alto, CA, USA) capable of supplying a certain drug concentration at a constant rate was developed and used in multiple animal species as small as mice [17]. This can be achieved by implantation at multiple anatomical sites for long-term drug delivery, generally over a 2- to 4-week period in the case of rats [18,19]. However, its osmotic action is only suitable for water-soluble drugs, and the cost of these devices is high [20]. In addition, when the amount of drug in the drug sac decreases, the elasticity and drug injection pressure decrease; thus, there is a disadvantage in that the drug injection amount is not maintained constant.

Herein, we propose a long-term epidural administration method for a lumbar stenosis model. This work highlights a standardized method to evaluate long-term administration of drugs that can be continuously injected into the rat epidural spaces.

## 2. Materials and Methods

### 2.1. Rat Spinal Stenosis Model

Male Sprague–Dawley rats (7 weeks of age, 230–250 g, Daehan Bio Link, Chungju, Republic of Korea) were separately housed in cages at a mean temperature of 25 °C and 50% humidity with a 12 h light/dark cycle. All experimental agreements were approved by the Jaseng Animal Care and Use Committee (JSR-2023-01-005-A). The surgical procedure for establishing a spinal stenosis model has been previously described in detail [21]. Briefly, a silicone block with a hardness of 80 kPa (SH5180U, KCC Corporation, Seoul, Republic of Korea) was inserted into the L4 epidural space without laminectomy. The rats were sutured with Surgicel^®^ absorbable hemostat (Johnson & Johnson, Arlington, TX, USA). After anesthesia was released, 40 mg/kg of cefazolin sodium (Cefazolin, Chong-Kun-Dang Pharm., Seoul, Republic of Korea) was injected intramuscularly and 10 mg/kg of acetaminophen syrup (Janssen Pharmaceuticals, Titusville, NJ, USA) was orally administered. Rats (n = 6 per group) were sacrificed 3 weeks after silicone implantation.

### 2.2. Intrathecal Catheter with Drug Delivery System

Intrathecal catheters were purchased from Instech Laboratories, Inc., Plymouth Meeting, PA, USA. We used 12 cm long polyurethane tubing [inside diameter (ID) 0.13 mm; outside diameter (OD) 0.25 mm] as an intrathecal catheter. The drug delivery device purchased from Instech Laboratories consists of a button, cap, and injector required to connect with a syringe. Bone cement (DePuy CMW™ 3 GENTAMICIN Bone Cement (Dupuy synthes, Leeds, UK) was used to prevent drug leakage from the intervertebral space after catheterization and induction of stenosis.

### 2.3. Histological Analyses

Transcardiac perfusion with 0.9% normal saline (Sigma-Aldrich, St. Louis, MO, USA) and 4% paraformaldehyde (Biosesang, Seongnam, Republic of Korea) was performed 3 weeks after catheter implantation in lumbar spinal stenosis (LSS) rats. The 2 cm spinal cord sample from the T10 level (start of catheter insertion) was post-fixed overnight in 4% paraformaldehyde at 4 °C and cryoprotected in 30% sucrose in PBS for 3 days. Tissues were sectioned in the sagittal plane into 16 µm sections using a cryo-microtome (CM1520, Leica Biosystems, Nussloch, Germany). The sectioned tissues were incubated with rabbit anti-monocytes/macrophages (1:500; Abcam, Cambridge, UK) at 4 °C overnight. After washing thrice with PBS, Alexa Fluor 488-labeled goat anti-mouse secondary antibody (Abcam) was added at 1:200 in 2% normal goat serum for 2 h. The stained tissue was imaged using a confocal microscope (Eclipse C2 Plus, Nikon, Tokyo, Japan).

### 2.4. Confirmation for Epidural Catheterization

First, Evans blue staining was performed to confirm the correct catheter positioning. Rats were anesthetized with 2–3% isoflurane gas, and 150 µL of 1% Evans blue was injected into the epidural space through the implanted catheter. Immediately after epidural injection, the tissues were removed for macroscopic analysis using an inverted microscope (Nikon, Tokyo, Japan). In addition, 0.1 mL of 2% lidocaine was injected to confirm paralysis of the rat’s hind limbs, which confirmed successful epidural transplantation.

### 2.5. Locomotor Function Assays

We used three tests to assess locomotor function after implanting catheter and inducing stenosis: the Von Frey test; the Basso, Beattie, and Bresnahan (BBB) scale; and the horizontal ladder test. The Von Frey test was used to measure the rats’ response to pain. The rats were kept in acrylic cages for 15 min before their ability to adapt was measured. The latency of paw withdrawal in response to mechanical stimulation applied to the center of both hind paws using a Von Frey filament (Ugo Basile, Varese, Italy) was measured. An avoidance response by lifting, whipping, licking, or running off the foot during stimulation was considered a positive reaction, and the average value of ≥3 measurements was used. The BBB scale was expressed as a score from 0 to 21 points (0: no hindlimb movement; 21: normal hindlimb movement). Two independent observers analyzed hindlimb motion in an open field (a cylindrical acrylic box; 90 cm diameter, 15 cm height) for 4 min. The average values were used. The ladder walking test was used to assess the ability of the rats to maintain balance. All rats walked on a metal runway (2.5 cm between grids) from left to right three times, and their movements were recorded with a digital camcorder. The score was calculated as follows: ladder score (%) = erroneous steps of the hind limb/total steps of the hind limb × 100.

Locomotor functions were examined in each group every 7 days until sacrifice. All locomotor tests were recorded using a digital camera and performed by two observers who were blinded to the treatments.

### 2.6. Statistical Analyses

All numerical data are expressed as the mean ± standard deviation. Prism software (GraphPad, San Diego, CA, USA) was used for all the analyses. A one-way analysis of variance (ANOVA) with Tukey’s post hoc test was used to confirm significant differences between the sham and LSS groups. Statistical significance was set at *p* < 0.05.

## 3. Results

### 3.1. Preparation for Epidural Catheter Insertion and Stenosis Induction in Rats

To implant the catheter into the epidural space in rats, we first corrected the catheter according to the vertical length of the spine in 7- or 8-week-old male Sprague–Dawley rats. The catheter was divided into three parts (length 1, silicone collar, and length 2), length 1 not inserted into the epidural space and length 2 inserted into the epidural space, centering on the silicone collar (Figure 1A). The original lengths of lengths 1 and 2 were 6 and 12 cm, respectively. The length 1 part was cut to 3 cm, and the length 2 part was cut to 7 cm. The purchased drug injection device consisted of an injection port lid, a drug inlet (one channel), and an injector to connect a 1 mL syringe. (Figure 1B). The modified catheter was connected to the drug delivery device before the catheter was implanted into the animal (Figure 1C). Sterile surgical instruments (including forceps, poles, scissors, retractors, and hemostatic forceps) were used for catheter implantation and spinal stenosis induction; surgicel and spongostan were used as auxiliary tools for the surgical procedure to control hemostasis (Figure 1D). The following image is that of a silicon block which has been used previously to induce moderate spinal stenosis; the hardness is 80 kpa, and the size is 4 × 1 mm^2^. A silicon block with a hardness of 80 kpa was prepared by placing it in a petri dish with ethanol. Figure 1E shows the bone cement that was made by mixing the two components. It should be mixed immediately prior to placement to prevent drug leakage after implanting the catheter and silicone.

### 3.2. Implantation of a Drug Delivery Device into the C2 Level of Rats

A rat was placed in the prone position, de-furred at the C1 to L6 level, and covered with povidone-iodine (betadine). The drug delivery device was inserted at the C2 level so that the rat’s hands and feet did not touch, and the modified catheter was inserted at the T10 level. Silicone was implanted to induce spinal canal stenosis at the L4 level (Figure 2A). Skin incisions were made at the C2 and T10 levels before catheter insertion using a scalpel with blade of approximately 1.5–2 cm each (Figure 2B). Since the length 2 part of the catheter had an inner diameter of 0.13 mm and was very flexible, a plastic transparent spinal needle cover was used as a guide to insert the catheter under the skin. After inserting the spinal needle cover through the incised skin at the C2 and T10 levels, a catheter connected to the drug delivery device was inserted into the cover. After insertion, only the guide cover was removed (Figure 2C).

### 3.3. Catheter Placement in the Epidural Space Starting at the T10 Level

A series of images shows the procedure for implanting the drug delivery device above the C2 level (Figure 3A). After placing the device inside the incised skin at the C2 level, the incised skin was sutured with the device using the purse-string suture method with a 3-0 prolene suture. This suturing method is important to protect against the risk of infection. Before inserting the catheter into the epidural space, it was firmly fixed by suturing the front and back of the silicone collar to the muscle (Figure 3B). In the catheter image described in Figure 1A, the length 1 portion with an outer diameter of 0.6 mm was not inserted epidurally. Next, we completely removed the muscles attached to the vertebrae at the T9 and T10 level using a rongeur. As shown in Figure 3C, the surrounding muscles were completely removed until only the white vertebrae were visible, and the catheter was inserted through the space between the T9 and T10 levels without laminectomy (Appendix A). The reason laminectomy was not performed was to prevent drug leakage from the epidural space, and the epidural space for catheter implantation was sufficiently secured by grasping the spinous process at the T9 level with a tooth forceps and slightly opening it.

### 3.4. Prevention of Drug Leakage Using Bone Cement and Animal Care Procedures after Surgery

After the catheter was inserted into the epidural space through the hole between the vertebrae at the T9 and T10 levels, the hole was closed with bone cement to prevent drug leakage. Then suturing was performed using 5-0 sutures in the reverse order of incision (Figure 4A). In addition, silicon implantation was performed in the spinal canal to induce spinal stenosis. In the same way as in the previous spinal stenosis model fabrication method [21], silicone with a size of 4 × 1 mm^2^ and hardness of 80 kpa was inserted at the L4 level through the space between L5 and L4 without performing laminectomy (Appendix A). After insertion, bone cement was placed to block the hole to prevent drug leakage from the space between L4 and L5. After suturing in the reverse order of the incision using a 5-0 suture, betadine was applied to the skin for disinfection (Figure 4B). Finally, we injected antibiotics intramuscularly and administered analgesics orally (Figure 4C).

### 3.5. Evaluation of Epidural Catheter Placement by Evans Blue, Lidocaine Injection, Immunohistochemistry, and Functional Assessments

We evaluated whether the catheter was correctly implanted into the epidural space through the drug injection port after epidural catheter insertion in rat with spinal stenosis. The rat was euthanized immediately after Evans blue injection and axially sectioned at the L4 level. Figure 5A shows that the epidural space was specifically stained with Evans blue. Therefore, this result confirmed that the catheter was correctly positioned in the epidural space.

Next, we observed whether temporary paraplegic symptoms appeared after injecting 100 µL of 2% lidocaine through the drug inlet. The rat walked normally before the lidocaine injection; however, transient paraplegic symptoms were observed after lidocaine injection (Appendix A). Figure 5B shows that the hind paws were facing upwards, as in a rat with spinal cord injury.

Furthermore, we examined whether the induction of inflammation in the spinal cord tissue following catheter epidural implantation was confirmed using CD68 staining, a macrophage-specific marker (Figure 5C). The macrophages stained with CD68 were not observed in the spinal cord of the catheter group similar to the sham group. CD68-stained macrophages were also not observed in the group implanted with a catheter for spinal stenosis. This is because the tissue used for histological analysis was a 2 cm spinal cord tissue sample at the catheterized T10 level and not the silicone implanted at the L4 level.

Lastly, gait changes were assessed weekly for up to 3 weeks using the BBB scale, ladder scoring, and the Von Frey test. The catheter group had normal gaits similar to the average score of the sham group until 3 weeks. (Figure 5D). The relatively low BBB scale scores for the rats with catheterization for spinal stenosis imply that catheter implantation did not cause gait disturbance and gait impairment due to stenosis. Ladder scoring showed no significant difference for the mean value of the sham group in the catheter group until 3 weeks. The increase in mis-stepping from the ladder scores were confirmed at 3 weeks after induction of spinal stenosis (Figure 5E). Further, Von Frey tests for 3 weeks revealed that no significant difference in left latency was observed between the sham and catheter groups, although left latency was significantly reduced for up to 3 weeks in the catheter implantation group after induction of stenosis compared to the catheter group (Figure 5F).

## 4. Discussion

Epidural injections are commonly used to treat chronic pain, a major public health issue that affects millions of people worldwide [22]. However, there is a lack of standardized methods for long-term epidural administration that can be consistently used across different preclinical research studies. Additionally, standardizing the method of epidural administration can help minimize variability in results and reduce the risk of adverse events in preclinical research. By establishing a consistent and well-established method, researchers can be confident that the results of their studies are valid and can be compared to those of other studies.

Herein, we describe a comprehensive and standardized protocol for epidural administration in LSS rats using a commercially available device. We conducted various trial-and-error experiments to set up the protocols. First, to minimize the pressure on the spinal cord due to the catheter implanted in the epidural space, we purchased an intrathecal catheter with an outer diameter of 0.25 mm and a size of 32 gauge, originally used for implantation in the intrathecal space. Most of the existing catheters implanted for epidural injection are PE-10, made of polyethylene (PE) with an outer diameter of 0.61 mm. However, previous studies have reported that polyurethane material could minimize nerve damage compared to PE material and that the smaller the size of the catheter, the less pressure applied to the spinal cord. In addition, we modified the purchased intrathecal catheter according to the spinal length of Sprague–Dawley rats, which weighed approximately 250 g at 8 weeks. As explained in the results section, intrathecal catheters were standardized at 7 cm from the center of the silicone collar to the epidural space and 3 cm from the non-inserted portion. The reason for setting such a length was that if the length of the second part is too short, the inserted catheter may come out of the epidural space and the length of the first part that could not be inserted into the epidural space could not be adjusted to <3 cm owing to the problem of excessive injection pressure. Therefore, the catheter could be inserted into the epidural space from T10, and the length of the catheter to reach the L4 level in the epidural space was 7 cm.

For catheter fixation, a reference was made to secure the catheter in place using adjacent muscle fascia [16]. We sutured the implanted catheter with the muscle from above and below, centering on the silicone collar to secure the implanted catheter. A drug administration device for daily administration from the outside into the epidural space for a long period may cause infection. To minimize this, the inlet was sutured to the skin using a purse-string suture method [23]. By observing the condition of the drug inlet connected to the epidural space for 3 weeks, we confirmed that there were no problems such as clogging of the inlet, inflammation, or infection. Finally, after evaluating that the catheter was accurately implanted in the epidural space and that the tip of the catheter was located at the L4 level, which we targeted, 150 µL of 1% Evans blue was injected into the drug inlet, and the rats were sacrificed immediately. Tissue sectioning was performed in the axial view. The epidural space was stained blue because of the Evans blue staining. In addition, as a result of injecting 100 µL of 2% lidocaine into the drug inlet, a temporary state of lower extremity paralysis was observed for approximately 5–10 min. Finally, we sacrificed the rats 3 weeks after implantation to check that the catheter connection was not separated from drug delivery device and that it was correctly positioned at the implanted site (Appendix A). As a result of checking, the drug infusion device and catheter were not separated, and the catheter was correctly positioned without departing from the place where it was implanted.

Experimentally, the rat epidural administration method is a valuable tool for researchers studying pain and anesthesia in rats and has been shown to provide effective and targeted delivery of drugs to the spinal cord and surrounding tissues. Our protocol may be useful for evaluating the effect of epidural catheterization in rats for long-term pharmacological studies, including its effects on motor function and inflammatory markers. It can be used in preclinical studies related to spinal models such as spinal cord injury, spinal cord compression, and various disc diseases. Furthermore, this method will increase the accuracy and reliability of the results, allowing for more robust comparisons between studies and advancing our understanding of the effects of epidural injections for the treatment of chronic pain.

## 5. Conclusions

We proposed a new method for rat epidural catheterization and long-term administration to study the effects of drugs for pain management and to develop novel pharmacological therapies for pain relief.

## Figures and Tables

**Figure 1 biomedicines-11-01390-f001:**
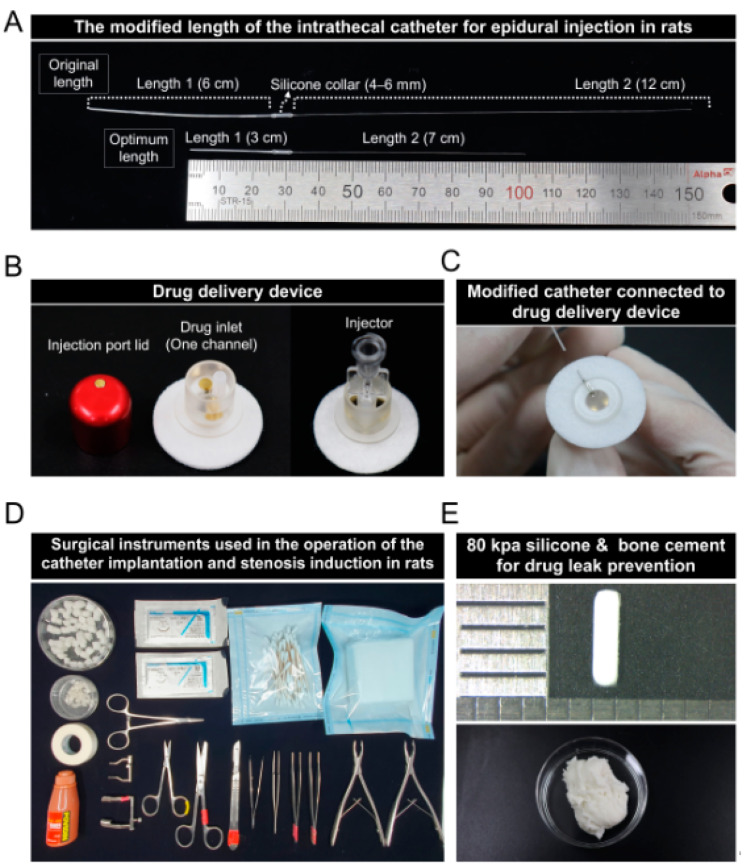
A modified delivery catheter for epidural injections in lumbar spinal stenosis rats. (**A**) A 32-gauge intrathecal catheter was modified to fit the spine length of 7–8-week-old Sprague–Dawley rats. (**B**) The drug delivery device consists of three components: injection port lid, drug inlet port, and injector for delivering drugs externally through the inserted catheter into the epidural space. (**C**) The captured image shows connection of the modified catheter to the drug delivery device. (**D**) Surgical tools including forceps, scissors, rongeurs, retractors, and supplies used for the catheter and silicone implants. (**E**) An 80 kPa silicone and bone cement prepared for spinal stenosis induction and drug leakage prevention.

**Figure 2 biomedicines-11-01390-f002:**
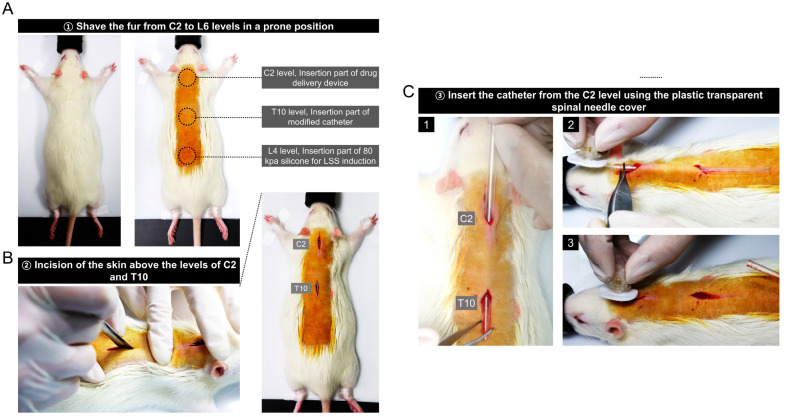
Insertion of the modified catheter through the incised skin above the level of C2 and T10 in a rat. (**A**) The rat lying prone on the operating table and the preoperative site is being marked. (**B**) The skin incision at the C2 and T10 levels using a scalpel blade. (**C**) Series images of catheterization using a transparent plastic spinal needle cover as a guide.

**Figure 3 biomedicines-11-01390-f003:**
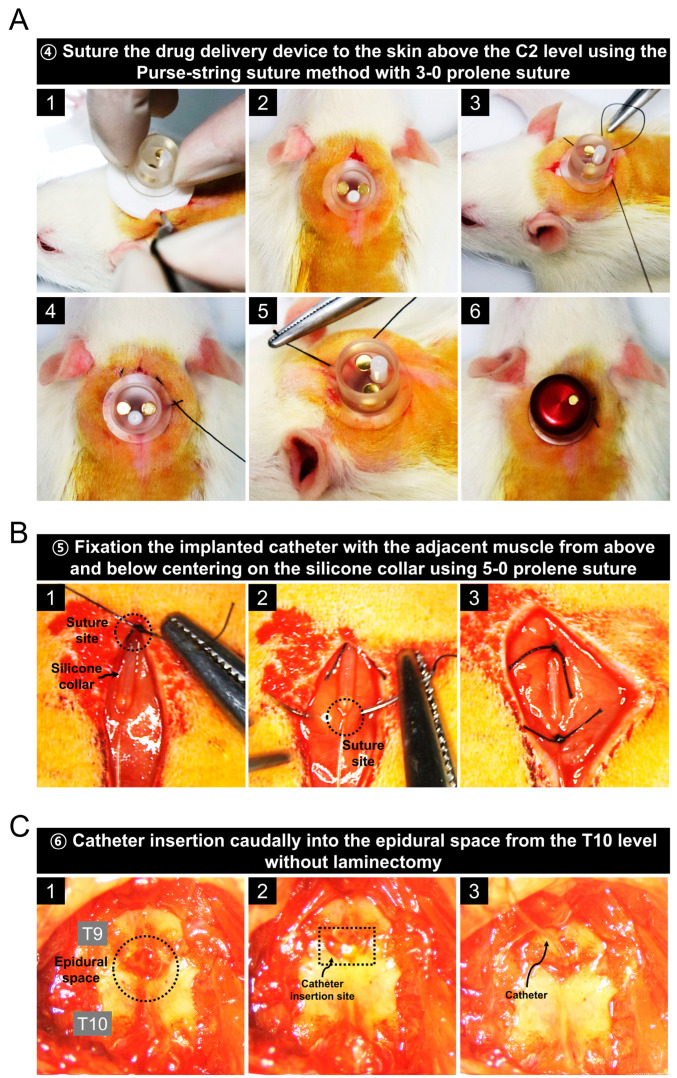
Procedure for implanting the drug delivery device at the C2 level and inserting a catheter into the epidural space between T9 and T10 levels in a rat. (**A**) Series images of drug delivery devise implantation with a 3-0 prolene suture and using the purse-string suture technique. (**B**) Images of the catheter fixation procedure that is not inserted into the epidural space. (**C**) Images related to the procedure for catheter insertion into the epidural space between the T9 and T10 levels without laminectomy.

**Figure 4 biomedicines-11-01390-f004:**
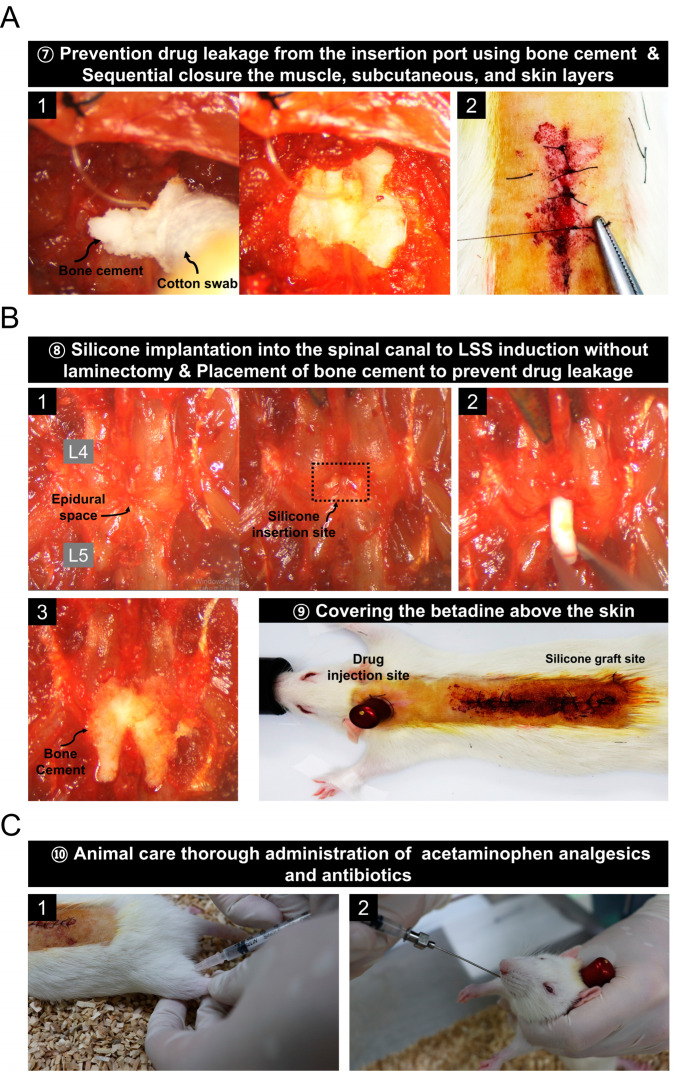
Procedure for preventing drug leakage using bone cement and creating an animal model for spinal canal stenosis. (**A**) Images of the bone cement implantation process after catheter insertion into the epidural space and sequential suturing using a 5-0 suture. (**B**) A series of images of the process for inducing spinal stenosis by implanting silicone into the spinal canal without laminectomy. (**C**) Post-operative care procedures.

**Figure 5 biomedicines-11-01390-f005:**
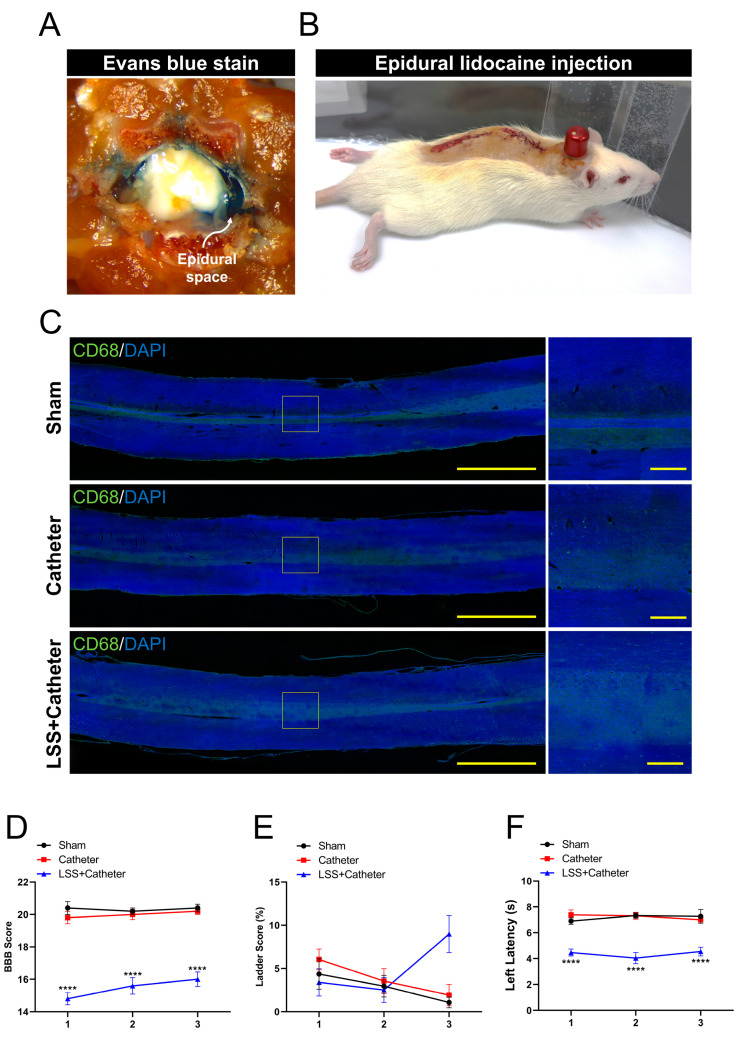
Evaluation of epidural administration using a drug delivery device after implanting a modified catheter into the epidural space in rats with spinal stenosis. (**A**) Representative image of Evans blue staining into the epidural space induced by an implanted catheter. (**B**) Image of a rat with transient paraplegia after lidocaine injection by epidural. (**C**) Immunohistochemical images of CD68^+^ macrophages at 3 weeks after catheter implant or catheter implant with stenosis induction, revealing no inflammation in the spinal cord tissue from catheter implantation. Locomotor function was measured every week for up to 3 weeks. (**D**) Basso, Beattie, and Bresnahan scale scores by open field test. (**E**) ladder scores by horizontal walking test. (**F**) Latency for left response by the Von Frey test. The data are expressed as the mean ± SEM and were analyzed via two-way analysis of variance with Tukey’s post hoc test. Significant differences are indicated as follows: **** *p* < 0.0001 vs. catheter group.

## Data Availability

Data is contained within the article and Appendix A.

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
