# Peer review of "Epidural Injection Method for Long-Term Pain Management in Rats with Spinal Stenosis"

_biomedicines, 2023, doi:10.3390/biomedicines11051390_

Round 1
Reviewer 1 Report
The manuscript written by Hong et al., is well organized and explains rationale, methods as well as results clearly. Hong et al., developed a standardized method for long-term epidural injections for evaluating the efficacy and safety of drugs in rats with lumbar spinal stenosis. Methods, results, and figures are described in detail and easy to follow. In the Discussion section, authors have compared their method with existing methods of epidural injections and demonstrated the efficacy of their novel method established herein. The conclusion of the study is well supported by data presented in the manuscript.
Author Response
Many thanks, We highly appreciate the effort and time devoted by reviewer while evaluating this manuscript
Reviewer 2 Report
Thank you for permitting me to review this manuscript
In this animal study authors described a new technique of long duration epidural catheter delivery used in rats
LINE 67-71 , these sentences are speculation and should be stated in discussion only
2.4 line 110 please add a legend for picture of confirmation of epidural catheter insertion figure 5A ,?
Since the authors state themselves that this sytem may be useful only for rats , this should also be stated in the title , therefore the discussion should more focus on rats especially in the introduction
Please describe why the drug delivery in C2 is important as I understand this is for the rats not touching it with their arms , this may be stated before in the method section and not in the discussion section in order to better understand the purpose
did the authors checked the absence of leak after device implementation
If the authors project the use of this system later in humans they should state it specifically in the discussion
Author Response

(The authors gave the same response as above.)

Reviewer 3 Report
The method is well-described. The authors should acknowledge it may be technically challenging and may not be reproducible in all laboratories.
Author Response
Many thanks, We highly appreciate the effort and time devoted by reviewers while evaluating this manuscript. As advised by the reviewer, the novel long-term epidural injection method proposed by us was confined to rats as a research background in the introduction section